# Refinement in the European Union: A Systematic Review

**DOI:** 10.3390/ani12233263

**Published:** 2022-11-23

**Authors:** Alina Díez-Solinska, Oscar Vegas, Garikoitz Azkona

**Affiliations:** Department of Basic Psychological Processes and Their Development, Euskal Herriko Unibertsitatea (UPV/EHU), Tolosa Hiribidea, 20018 Donostia, Spain

**Keywords:** 3Rs, refinement, financial support, European Union

## Abstract

**Simple Summary:**

More than 10 years have passed since the publication of Directive 2010/63/EU on the protection of animals used for scientific purposes based on replacement, reduction, and refinement (3Rs). These principles state that if animals have to be used in experiments, researchers should make every effort to replace them with non-sentient alternatives, reduce them to a minimum, and refine experiments and housing conditions so as to cause the minimum possible pain and distress. In this systematic review, we aimed to identify and summarize published advances in the refinement protocols made by European Union-based research groups from 2011 to 2021, and to determine whether or not said research was financially supported. Our results indicated that the majority of advances were related to improvements in experimental procedures for mice, and the research groups were mostly from universities and the United Kingdom. More than two thirds of the studies received financial support, mostly national. There is a clear willingness in the scientific community to improve the welfare of laboratory animals. However, we believe that more progress in refinement would have been made during these years if there had been more specific financial support available at both the national and European Union levels.

**Abstract:**

Refining experiments and housing conditions so as to cause the minimum possible pain and distress is one of the three principles (3Rs) on which Directive 2010/63/EU is based. In this systematic review, we aimed to identify and summarize published advances in the refinement protocols made by European Union-based research groups from 2011 to 2021, and to determine whether or not said research was supported by European or national grants. We included 48 articles, the majority of which were related to improvements in experimental procedures (37/77.1%) for mice (26/54.2%) and were written by research groups belonging to universities (36/57.1%) and from the United Kingdom (21/33.9%). More than two thirds (35/72.9%) of the studies received financial support, 26 (mostly British) at a national level and 8 at a European level. Our results indicated a clear willingness among the scientific community to improve the welfare of laboratory animals, as although funding was not always available or was not specifically granted for this purpose, studies were published nonetheless. However, in addition to institutional support based on legislation, more financial support is needed. We believe that more progress would have been made in refinement during these years if there had been more specific financial support available at both the national and European Union levels since our data suggest that countries investing in refinement have the greatest productivity in successfully publishing refinements.

## 1. Introduction

In the European Union (EU), the protection of animals used for scientific purposes is regulated by Directive 2010/63/EU [1]. In order to harmonize standards across the EU, member states were required to transpose the Directive into national legislation and most of them did so during 2013 [2]. In 2019, the Directive was amended to increase transparency [3]. This Directive is considered an essential piece of legislation with which anybody who carries out fundamental biological research and preclinical development potentially involving live cyclostomes, cephalopods, and/or vertebrate animals must be familiar [4].

Overall, the Directive promotes both animal welfare and high-quality scientific research and establishes one of the most progressive and stringent mandatory lab animal protection frameworks in the world [4]. It was drafted with four very clear fundamental principles in mind. First is the recognition that the ultimate goal is to replace the use of animals. Second is the acknowledgment that animals, including non-human primates, are still needed today. Third is the acceptance that animals have intrinsic value in themselves and must be respected. Fourth is the agreement that the principle of the Three Rs (3Rs) is the key to ensuring more humane and better science [5].

The principle of the 3Rs—replacement, reduction, and refinement [6]—is the cornerstone of the Directive. These principles state that if animals have to be used in experiments, researchers should make every effort to replace them with non-sentient alternatives, reduce them to a minimum, and refine experiments and housing conditions so as to cause the minimum possible pain and distress. Thus, the 3Rs concept is both a framework designed to minimize the use and suffering of animals (harm to the animal) and a means to support high-quality science and translation (benefit to society). The conflict between these two aims is usually resolved on a case-by-case basis by weighing up the harm to the animals involved and the benefits of the research, or by prioritizing the experience of the animals (i.e., refinement) over reduction [7]. 

Whereas there is a greater consensus regarding the replacement and reduction principles, the implementation of the refinement principle has caused the greatest controversy. Refinement is an ongoing process that requires input from all those involved in the use of experimental animals [8] and covers all animal and human interactions throughout the entire life of the animal. It is not limited solely to experimental procedures, but rather also encompasses the transport, husbandry, and euthanasia of animals [9]. Recently, we observed that people who work with laboratory animals are clearly aware of this and show great sensitivity to their well-being [10,11]. Moreover, perceived animal stress/pain negatively affects the professional quality of life of people working with laboratory animals [12].

In this systematic review, we aimed to identify and summarize published advances in refinement protocols made by EU-based research groups from 2011 to 2021 and to determine whether or not the research was supported by European or national grants. 

## 2. Materials and Methods

This systematic review is reported in accordance with the Preferred Reporting Items for Systematic Reviews and Meta-Analysis (PRISMA statement) flowsheet. 

### 2.1. Search Strategy

Web of Science and PubMed were chosen for the search. The search was carried out on 13 May 2022. As the main aim of the study was to examine the number of original publications developing and/or improving a refinement technique (considering Russell and Burch’s 3Rs) in laboratory animal research, the following search terms were used in both databases: (3 Rs OR 3 R OR 3R OR 3Rs) AND (refinement) AND (animal*) AND (techniqu* OR strateg*). The filters included were publication year (from 2011 to 2021), document type (article), and language (English). In the Web of Science database, the countries/regions filter was also applied (countries from the European Union), whereas PubMed articles from outside the European Union were excluded manually. Since our goal was to select articles refining animal techniques, we also used Web of Science filters to exclude human research. We comprehensively searched for published full-text studies. The study selection was performed by A.D.-S., and G.A., who independently examined the full texts of potentially relevant studies and applied the eligibility criteria in order to select, by consensus, those studies to be included. The information extracted from the articles included the title, authors, year of publication, DOI, animals in which the refinement technique was implemented, country in which the study was performed, the kind of procedure that was refined and a brief description of it, the institution to which the authors were affiliated, whether or not the article had received any kind of financial support, and which type of institution funded the studies. These data are provided in Appendix A.

### 2.2. Eligibility Criteria 

Original articles were deemed eligible if they met the following criteria:-The articles were published from 2011 to 2021 (both years included).-At least one of the authors of the article belonged to an institution/research center/university (public or private) from the European Union.-The studies were not classified as reviews, meta-analyses, or reports, nor were they guidelines, protocols, surveys, or ethical discussions for research reflection. -The main topic of the studies was the improvement or development of a refinement technique, not a replacement or reduction method, even when this was linked to the 3Rs. -The studies were published in English.

### 2.3. Categorization of Refinement Procedures

We categorized studies based on whether they proposed improvements in experimental procedures, husbandry, transport, or euthanasia. Within these categories, we defined sub-categories with the aim of grouping studies in accordance with the common characteristics of the procedures.

### 2.4. Statistical Analysis 

Frequency (%) statistics were used to describe the sample.

## 3. Results

### 3.1. Study Selection 

We systematically searched for references related to refinement procedures in laboratory animals. A total of 384 references were identified by electronic search; 338 full-text studies were evaluated in accordance with the eligibility criteria and 290 were excluded. Finally, 48 studies [13,14,15,16,17,18,19,20,21,22,23,24,25,26,27,28,29,30,31,32,33,34,35,36,37,38,39,40,41,42,43,44,45,46,47,48,49,50,51,52,53,54,55,56,57,58,59,60] complied with all the established eligibility criteria and were included in the study (Figure 1).

### 3.2. Refinement Procedures by Category

First, we categorized the different refinement procedures into previously defined categories (Table 1). Experimental procedure was the category into which most studies fell (37/77.1%), followed by husbandry (10/20.8%), with only one study being categorized as refinement in transport (2.1%). We did not find any studies refining euthanasia protocols.

Of the experimental procedure sub-categories, sampling encompassed the most studies (10/28.6%), followed by analgesia (4/10.8%) and animal training (4/10.8%). Regarding husbandry, welfare assessment in the animals’ natural environment and social housing were the sub-categories containing the highest number of published studies (3/30% each).

### 3.3. Refinement Procedures by Animal Species, Institution, and Country

More than half of the published studies described refinement procedures for mice and just over twenty percent did so for rats, meaning that most of the protocols described were for rodents. The remaining procedures were described as refinements for other mammals such as macaques, dogs, or pigs, as well as for fish and birds (Figure 2a). 

More than half of the research groups belonged to universities, just over a quarter belonged to research institutes, and just under ten percent were from private companies (Figure 2b). Moreover, 14 (29.2%) of the studies were collaborations between different institutions: private company and research institute (1); university and private company (2); university, a private company, and research institute (1); university and research institute (9); and university and zoo (1). 

In terms of country, most research groups were based in the United Kingdom (UK), followed by Germany (Figure 2c). Of the 48 articles selected, 13 (27.1%) were collaborations between groups from several countries: Austria and Germany (3); Belgium and the UK (2); Czech Republic, Denmark, and Sweden (1); Denmark and Sweden (1); France and Norway (1); Germany and Spain (1); Germany, the UK, and Spain (1); Hungary, Finland, and the UK (1); the UK, Australia, and New Zealand (1); and the UK, Australia, and South Africa (1). 

### 3.4. Financial Support for the Studies

More than two-thirds (35/72.9%) of the studies received financial support, 26 in the form of national funding, 7 from their own institution, and 2 from private foundations. Moreover, 8 received European funding: (1) COST Action (the Behavioral Management and Training of Laboratory non-human Primates and Large Laboratory Animals—PRIMTRAIN) [61], (1) the European Regional Development Fund, (2) the European Research Council (ERC), (1) the EU Integrated Project (Xenome), (1) the Innovative Medicine Initiative (IMI), (1) the Sex’NPerch program by the European Maritime and Fisheries Fund, and (1) the Seven Framework Program (FP7-HEALTH-MITOTARGET). Of the remaining studies, 8 (16.7%) received no financial support and 6 (12.5%) did not specify (Figure 3). Appendix A shows the number of un-funded and funded articles per country.

Studies carried out by the UK research groups obtained the most funding at both the EU and the national level. Of the 21 UK articles published, 18 (85.7%) were funded, 16 (76.2%) of which were funded nationally. Of these 16, half (8/50%) were partially or fully funded by the National Center for the Replacement, Refinement, and Reduction of Animals in Research (NC3Rs), and a quarter (4/25%) were also funded at a European level. The remaining two studies were conducted by groups working in private companies and were funded by their own institution. Of the 16 studies published by German groups, 11 (43.75%) received funding, 7 (63.6%) at a national level. Of these, one also received European funding and another one received private funding. The rest (4/36.4%) received institutional funding. One of these institutions is the Charité 3^R^ of the Universitätsmedizin Berlin, which actively promotes the 3Rs principle in biomedical research and education. One Swedish study received funding from a private foundation that promotes scientific research against painful animal experiments (Torvald and Britta Gahlin’s foundation).

## 4. Discussion 

An important component of good scientific practice is to reduce the suffering of laboratory animals through refinement techniques. In our systematic review, we identified 48 studies conducted by EU-based research groups between 2011 and 2021 that aimed to improve the welfare of animals used in research. We chose these 10 years because they correspond to the decade following the publication of Directive 2010/63/EU, which clearly promotes refinement [1]. 

During these ten years, tissue sampling improvements have been described to minimize the stress and pain associated with this procedure. Many of these methods are non-invasive and do not require great technical skill, thus reducing both the stress on the experimenter during handling and the harm to the animal. Similarly, other studies have sought to improve analgesia protocols by refining drug combinations or administration routes. The use of training both to condition and to habituate animals to a procedure is also worth noting. Although time consuming, training is a very good strategy for reducing animal stress and discomfort. Improvements have also been described in husbandry, with one area of focus being social housing. Many of the animals used in biomedical research belong to social species, and Directive 2010/63/EU recommends their group housing [1]. Technological advances are increasingly enabling animals to be monitored in their home cage, thereby reducing the stress associated with interaction with humans and improving their welfare. A COST (European Cooperation in Science and Technology) action is currently underway for this purpose [62].

Recent statistics on animal use in the EU indicate that mice are the most commonly used animals [63,64,65], and we observed the same trend in our review; more than half of the selected articles described refinements in mouse protocols. Regarding the origin of the research groups, in terms of institution and country, our results follow the same trend observed in the biomedical area [66], with university-based research groups from the UK being the ones with the most publications to their name.

Scientists are currently working to produce valid data on measuring and improving all laboratory animals’ welfare. In addition to the COST actions described above [61,62], the Eurogroup for Animal Welfare, a lobbying organization, is working to implement refinement methods in research [67]. However, we observed that there were only a few studies funded by European entities, and only one was specific to laboratory animals [61]. The rest of the European grants were oriented toward biomedical research. By country, the UK is the leading national funder of projects. It should be noted that the NC3Rs funded half of all the UK projects during this period. Other European countries also have centers dedicated to the achievement of the 3Rs, and a list of these can be found on the Norecopa webpage [68]. Our results show that specific funding for the achievement of the third R during this decade was far from substantial. We should not forget that scientists have been constantly asking for more public resources and interdisciplinary teams to solve the quandary of how to strike a balance between animal welfare and science [69].

The present study has certain limitations. First, since the word refinement can be applied to many scientific fields, the search was restricted to articles that also mentioned either animals or the 3Rs. In this sense, we were unable to identify some types of articles, such as, for example, strategies to reduce singly housed male mice [70], as well as others in which the authors did not identify the term refinement as a keyword [71]. Our strategy also has the possibility of underestimating the scope of refinements because some refinements are often published as part of the scientific work that animals are used in. Furthermore, our search would not have identified the refinements developed by research groups or animal facilities and implemented in their centers that were not published. In this sense, the use of platforms dedicated to the 3Rs may be a good tool to collect and disseminate these protocols, as it is sometimes difficult to publish them in indexed journals. The Animal Welfare Institute’s website contains protocols and scientific papers describing methods for reducing or eliminating pain, stress, and discomfort for animals, not only during experimental procedures but also in relation to their daily social and physical environment [72]. Our search did not include patented protocols such as, for example, the project “HaPILLness-Voluntary oral dosing in rodents”, which replaces oral gavage with voluntary dosing [73]. Finally, the results of the funded projects completed in 2021 may still be under preparation or under consideration.

Overall, our findings show that, in recent years, advances have been made in the refinement of procedures using laboratory animals. Currently, there are refined protocols that are used on a daily basis in many animal facilities, such as administering substances with sweetened condensed milk [74] or transporting mice through a plastic tube instead of holding them by the tail [75]. Moreover, statistics indicate that protocols classified as severe have been decreasing slightly in recent years, by 1% per year [65]. However, we cannot forget that in order to be able to carry out experiments to refine a technique, in addition to a multidisciplinary team in which veterinarians must play an essential role [76], financial support is still necessary.

## 5. Conclusions

Our results indicate a clear willingness among the scientific community to improve the welfare of laboratory animals, as although funding was not always available, or was not specifically granted for this purpose, studies were published nonetheless. However, in addition to institutional support based on legislation, more financial support is needed. We believe that more progress would have been made in refinement during these years if there had been more specific financial support available at both the national and EU levels since our data suggest that countries investing in refinement have the greatest productivity in successfully publishing refinements. 

## Figures and Tables

**Figure 1 animals-12-03263-f001:**
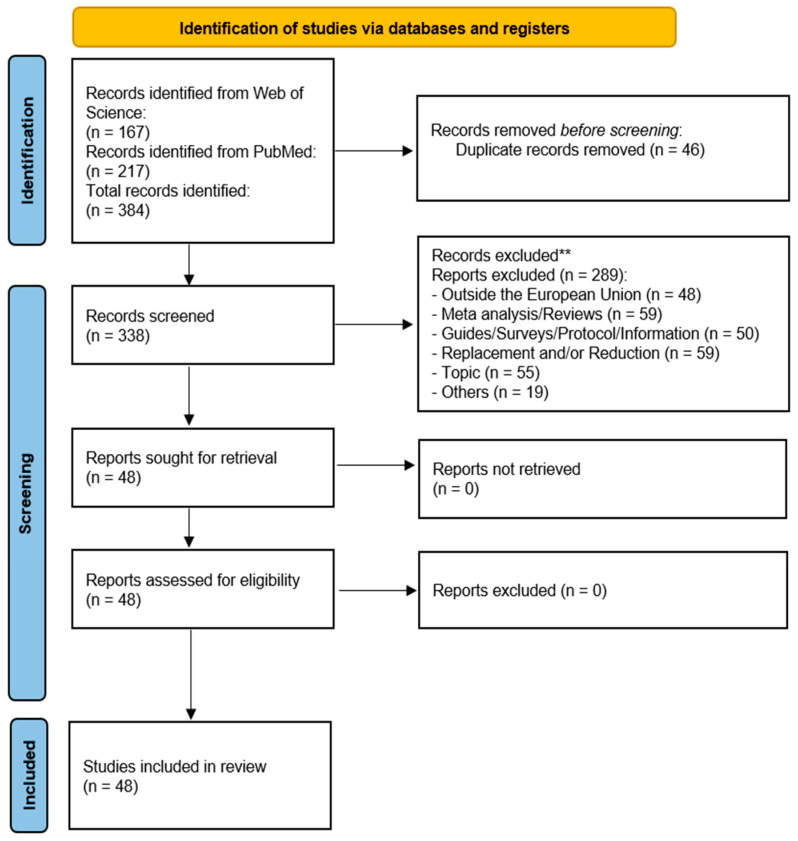
PRISMA flowchart of study selection.

**Figure 2 animals-12-03263-f002:**
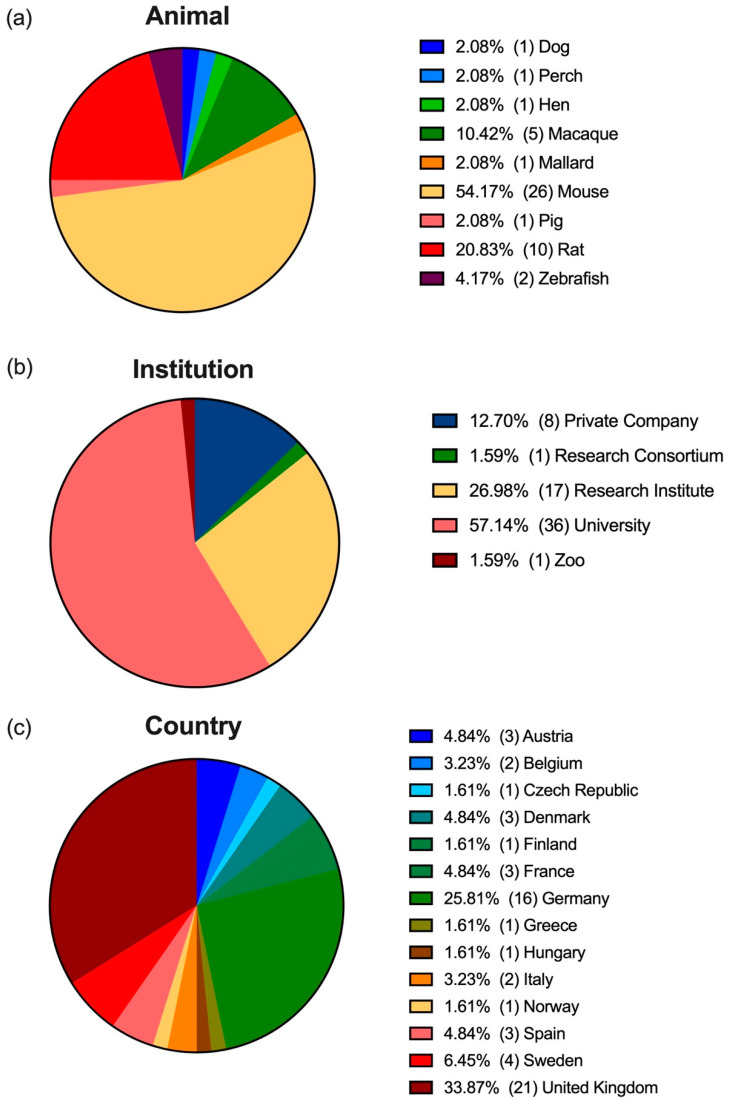
(**a**) Animal species for which the refinement protocols were described; (**b**) country; and (**c**) institution in which the research groups were working. Data are presented in percentages (total number).

**Figure 3 animals-12-03263-f003:**
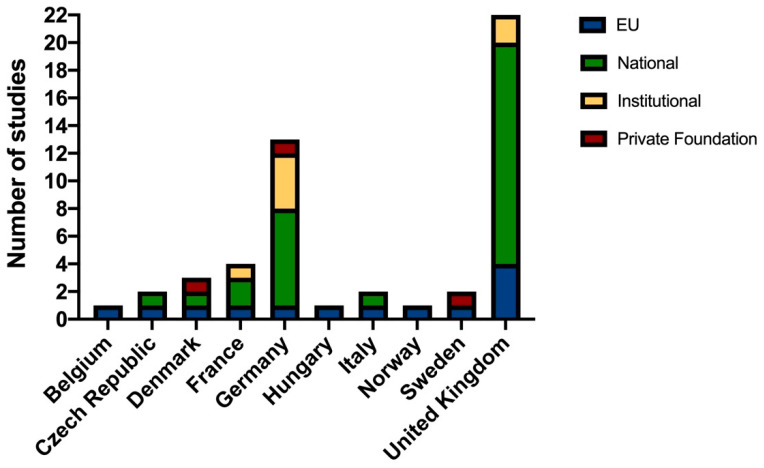
Financial support by country and organization. Data are presented in total numbers.

**Table 1 animals-12-03263-t001:** Classification of the studies by category and sub-category.

Category	Sub-Category	Brief Description	Reference
Experimental procedure	Analgesia	Buprenorphine is mixed with a desirable nut paste for mice to ensure its voluntary ingestion.	[13]
		Evaluation of different analgesic methods in order to improve their quality in surgical interventions.	[14]
		Refinement approach in the spinal cord injury model.	[15]
		Description of an intrathecal technique to administer morphine in order to minimize the interaction between this substance and the research itself.	[16]
	Behavioral Testing	Refinement of the Radial Arm Test to avoid animal handling and water and food deprivation.	[17]
		Reliable alternatives to the Water Maze Test in order to avoid the stress associated with deep water swimming.	[18]
	Diagnosis	Categorization of animals suffering from either hyperacusis or tinnitus through a model based on operant conditioning with positive reinforcement.	[19]
	Dosing	A new application of a device enabling the long-term enteral administration of drugs or nutritional supplementation.	[20]
		Syringe feeding as an alternative to the commonly used water bottles and oral gavage dosing in the administration of probiotics.	[21]
	Identification	A new tagging protocol to preserve animal welfare.	[22]
		A non-invasive technique for sexing in fish.	[23]
	Imaging	Thermographic imaging as an alternative method of examining tumor growth in a non-invasive way.	[24]
		Magnetic resonance imaging as a non-invasive imaging technique to detect early cancer onset and its development.	[25]
		The early detection of animal disease through imaging techniques.	[26]
		No need to use anesthesia for the procedure.	[27]
	Handling	A method to inject substances using a non-restrained technique.	[28]
	Others	Development of an implantable device for long-term Deep Brain Stimulation which provides mice with the freedom of movement.	[29]
		Reduction in exposure time to smoke and analyzing lung lobes separately in tobacco product testing.	[30]
		Refinement of a respiratory monitoring setup for an in vivo bioluminescence and fluorescence imaging device.	[31]
	Sampling	A non-invasive method for determining liver weight gain that reduces the number of animals needed and avoids the post-mortem technique.	[32]
		Blood micro-sampling from the ear capillary.	[33]
		Cutting feathers instead of plucking them from the animal’s skin in order to avoid this painful method.	[34]
		Non-invasive collection of murine hair follicle cells to avoid biopsies.	[35]
		Monitoring immunoglobulins via fecal pellets as a new non-invasive way to assess the immune response.	[36]
		Comparison of blood sampling techniques.	[37]
		Non-invasive ocular (tear) sampling for genetic ascertainment of transgenic mice.	[38]
		Refinement of sperm sampling techniques.	[39]
		Blood volume reduction in each sample.	[40]
		Swabbing the skin of non-anesthetized fish for DNA extraction.	[41]
	Surgery	Development of a surgical procedure to induce myocardial infarction in order to reduce mice mortality and distress.	[42]
		Refinement of a heterotopic cardiac transplantation model.	[43]
		Use of non-operated rather than sham-operated controls.	[44]
	Training	Cognitive enrichment allows mice to better lead with their home environment.	[45]
		Training dogs for gavage administration.	[46]
		An approach to alternative rewards (social stimuli instead of fluid droplets) in animal training.	[47]
		Refinement in training pigs prior to the oral glucose tolerance test.	[48]
	Welfare Assessment	Identification of pain and distress signals among macaques during experimental procedures in order to promote their wellbeing.	[49]
Husbandry	Breeding	Three techniques to refine the activation of CreERT2 in maternal mice.	[50]
	Environmental enrichment	An environmental enrichment tool to enhance animal welfare in large-scale mouse husbandry.	[51]
		A comparison of different environmental enrichment techniques and their advantageous effect on mice pups’ wellbeing and survival rate.	[52]
	Health	Optimization of a device to prevent mites from penetrating the skin of experimental hens.	[53]
	Social housing	A perforated transparent wall that divides the cage into two compartments and allows olfactory, acoustic, and visual communication between two mice but prevents fighting and injuries.	[54]
		Improvement in social housing in toxicology studies.	[55]
		An automated homecage system to register social alterations after pharmacological exposure.	[56]
	Welfare Assessment	A new tool to assess animal wellbeing, drawing on their natural behavior.	[57]
	Welfare Assessment	Use of home cage running wheels to obtain daily measurements of motor function in SOD1G93A mice, with minimal intervention.	[58]
	Welfare Assessment	Voluntary wheel running as an indication of distress suffered by mice in an experimental procedure.	[59]
Transport	Training	Reinforcement (positive and sometimes negative) to help monkeys acclimate more quickly to transport devices.	[60]

## Data Availability

All the data pertaining to the study will be made available upon request.

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
