# Peer review of "Refinement in the European Union: A Systematic Review"

_animals, 2022, doi:10.3390/ani12233263_

Round 1

Reviewer 1 Report

This review evaluates Refinement approaches in the EU with the intent to present the current landscape.  This was done via a deep literature search, relying on published work to present the source of funding.  

This is important work for policy-makers to understand the value of funding schemes dedicated to this work.  The authors should consider a few minor points to strengthen the manuscript.

In the abstract, the authors conclude with the statement that they believe more progress would be made with specific financial support.  While this is the logical conclusion, the authors should strengthen this with an evidence basis... e.g. their data suggests that countries investing in Refinement have the greatest productivity in successfully publishing refinements.

The timeframe selected to correspond with the directive is good.

Table 1 is dense but important, appreciated the summary of this in Figure 2.

Finally, in the discussion/conclusions:

When discussion the limitations of the searching only 'Refinement' literature, I would suggest to specifically emphasize that refinements are often published as part of the scientific work that animals are used in.  This search strategy has the possibility of underestimating the scope of refinements in this way that should be acknowledged.

The authors suggest "there is a willingness in the scientific community to improve welfare".  The authors point that some of this work is even done without funding support, which supports this comment. On the other hand, the authors cannot fully contextualize the impact of the funding (e.g. how much was spent to produce 48 papers, which may or may not be relatively modest in 10y)... This could be another limitation of evaluating refinement using this method.

Author Response

Dear Reviewer

Thank you very much for your comments, we have attached a point-by-point response to your suggestions.

Best regards

Author Response

(The authors gave the same response as above.)
